# Efficacy and Safety of Adenotonsillectomy for Pediatric Obstructive Sleep Apnea Across Various Age Groups: A Systematic Review

**DOI:** 10.3390/pediatric17040071

**Published:** 2025-06-25

**Authors:** Mohammed Halawani, Arwa Alsharif, Omar Ibrahim Alanazi, Baraa Awad, Abdulaziz Alsharif, Hawazen Alahmadi, Rayan Alqarni, Rahaf Mohammed Alhindi, Abdulmohsen H. Alanazi, Abdulmajeed Hassan Alshamrani

**Affiliations:** 1Pediatric ENT, King Abdullah Specialized Children’s Hospital (KASCH), Riyadh 11426, Saudi Arabia; halawanimo1@mngha.med.sa; 2Ministry of National Guard Health Affairs (MNGHA), Riyadh 11426, Saudi Arabia; 3King Abdullah International Medical Research Center, Riyadh 11426, Saudi Arabia; 4Department of Medicine and Surgery, Batterjee Medical College, Jeddah 21442, Saudi Arabia; 140215.rahaf@bmc.edu.sa; 5College of Medicine, King Saud University, Riyadh 11426, Saudi Arabia; alanazi323@ksau-hs.edu.sa; 6Department of Otolaryngology-Head & Neck Surgery, College of Medicine, King Saud bin Abdulaziz University for Health Sciences, Jeddah 21423, Saudi Arabia; awadba@ngha.med.sa; 7King Abdullah International Medical Research Center, Ministry of the National Guard—Health Affairs, Jeddah 22384, Saudi Arabia; 8Department of Medicine and Surgery, Vision College, Jeddah 23643, Saudi Arabia; 202313034@vision.edu.sa; 9Faculty of Medicine, Taibah University, Al-Madinah Almunawwarah 41477, Saudi Arabia; tu4354677@taibahu.edu.sa; 10College of Medicine, Imam Mohammad Ibn Saud Islamic University, Riyadh 13318, Saudi Arabia; 11College of Dentistry, King Saud University, Riyadh 11426, Saudi Arabia; 439100718@student.ksu.edu.sa (A.H.A.); 439100701@student.ksu.edu.sa (A.H.A.)

**Keywords:** obstructive sleep apnea, adenotonsillectomy, pediatric, age, AHI reduction, sleep quality, behavioral improvements, surgical complications

## Abstract

**Objectives**: To assess the safety and efficacy of adenotonsillectomy (AT) for treating uncomplicated pediatric obstructive sleep apnea (OSA) in children of different ages. **Methods**: A systematic search was conducted in four electronic databases, and 71 studies with a total of 9087 participants were included in the analysis. The studies were all before-and-after studies, cohort studies, and randomized controlled trials. Surgical results were analyzed according to age, disease severity, and follow-up duration. **Results**: Children younger than 7 years at the time of AT had a significantly greater decrease in disease severity, a greater decrease in hypoxemic burden, improved sleep quality, and improved cardiovascular function than children older than 7 years. Both cognitive and behavioral performance improved postoperatively, although these changes were more significantly associated with the duration of follow-up than with age at surgery. Notably, the rate of surgical complications was much greater in children under the age of 3. **Conclusions**: The current evidence indicates that AT is performed optimally between the ages of 3 and 7 years, offering the greatest chance of disease resolution and remission of associated conditions, balanced with a reduction in surgical risk. We highly recommend conducting high-quality randomized controlled trials to further inform the clinical guidelines for pediatric AT.

## 1. Introduction

This systematic review provides an overview of pediatric obstructive sleep apnea (OSA), a common disorder in childhood caused by upper airway dysfunction, resulting in partial or complete obstruction of the airway during sleep. This condition is associated with oxygen desaturation or disturbance during sleep [1], and its national prevalence is estimated at 1–5% among children, with some regional studies reporting even higher rates in school-aged children and those with risk factors such as obesity, craniofacial abnormalities, and allergic rhinitis [2]. The diagnosis of OSA is established primarily using overnight polysomnography (PSG), which monitors sleep parameters, including the apnea-hypopnea index (AHI), oxygen desaturation, and sleep structure. PSG is the gold standard for objectively diagnosing the severity of OSA in children [3]. Untreated pediatric OSA may lead to serious sleep disturbances, cognitive and behavioral dysfunction, and an elevated risk of cardiovascular complications [4]. These consequences significantly impact families and healthcare systems, both socioeconomically and in terms of public health [5].

Adenotonsillectomy (AT) is the first-line treatment for OSA in children, according to clinical recommendations [6]. However, multiple factors impact the surgical outcomes of AT, such as high body mass index (BMI), OSA severity, and comorbid conditions [7]. However, the underlying pathogenic mechanisms of OSA and its comorbidities may differ with age, as children undergo almost continuous growth and developmental changes. As a result, the age at which a child undergoes surgery may impact the efficacy of AT.

Despite the frequency of AT in pediatric practice, age-specific guidelines for surgical timing remain to be established. For instance, the American Academy of Pediatrics and other professional societies provide general recommendations for AT but do not differentiate treatment based on age stratification [8]. Consequently, the timing of AT often relies on individual clinician judgment rather than evidence-based consensus. Some retrospective studies suggest that older children may be at a greater risk for residual OSA following AT [9].

The mean age for AT is reported to be approximately 42.32 months (9–86 months) [10]; however, there is a lack of comprehensive, age-stratified data on AT outcomes in the region. Moreover, national clinical decisions continue to be made without standardized age-related outcome assessments [11].

Further studies that better analyze the age-dependency of AT efficacy would allow for optimizing decision-making in surgery and the consequent outcomes. To bridge this gap, we performed a systematic review to assess the effect of age on AT effectiveness in children with PSG-diagnosed, uncomplicated OSA. Multiple efficacy measures were used for assessment, including the severity of disease, hypoxemic load, quality of sleep, cardiovascular function, neurocognitive and behavioral outcomes, and surgical safety. The goal of this study is to present recommendations based on available evidence with a view toward optimizing the timing of AT in the pediatric population.

## 2. Materials and Methods

### 2.1. Search Strategy

The methodological approach for this systematic review was registered in PROSPERO (CRD42024574712) and conducted in accordance with the Preferred Reporting Items for Systematic Reviews and Meta-Analyses (PRISMA) guidelines (Appendix A and Appendix B) [12]. A systematic search was conducted in PubMed, Web of Science, and the Cochrane Library. The search strategy was established by M.H. and A.A. and approved by the research team. Studies in the literature pertaining to the effectiveness and safety of AT for the treatment of pediatric OSA were identified using a combination of Medical Subject Headings (MeSH) terms as follows “Tonsillectomy” OR “Adenoidectomy” AND “Apnea” OR “Hypopnea” OR Snoring” OR “Sleep Apnea” OR “Obstructive” AND “Pediatric” OR “Child.” Further articles were identified by checking the reference lists of the included studies.

### 2.2. Study Selection

Two independent reviewers screened all identified records and retrieved the full-text articles that satisfied the initial inclusion criteria. Differences in study selection were resolved through discussion.

#### 2.2.1. Inclusion Criteria

Studies were included if they met the following criteria: pediatric patients (<18 years) diagnosed with OSA using PSG based on widely accepted diagnostic guidelines at the time of publication and quantitative data available for outcomes of interest before and after surgery. In addition, the study should be published in English, and the study design should be case-control, cohort, randomized controlled trials (RCTs), or before-after studies.

#### 2.2.2. Exclusion Criteria

Studies were excluded if they focused exclusively on complicated pediatric populations, such as those with comorbidities that could independently affect post-operative outcomes. This included studies solely involving children with obesity (body mass index (BMI)≥ 95th percentile for age and sex), genetic syndromes affecting airway anatomy, craniofacial abnormalities, and neurological or psychiatric disorders impacting cognition or behavior. However, studies with mixed populations were included in this review.

#### 2.2.3. Screening and Data Extraction

Duplicate articles were removed following the primary search using Google Drive (Google: Mountain View, CA, USA) and Mendeley Desktop (Mendeley Ltd.: London, UK). The records were imported into the Rayyan [(https://www.rayyan.ai/) accessed on 23 September 2024] database for title and abstract screening by three independent reviewers (M.H., B.A., and A.A.). The full texts of the studies identified as potentially eligible were subsequently reviewed by four other independent reviewers (R.M.A., O.I.A., A.A., and H.A.) for final inclusion [13]. Discrepancies were discussed with R.A.A., A.H.A., A.H.A., and the study team and resolved. Study limits were applied for data extraction into an Excel sheet, including the study title, author(s), country, year of publication, journal, study design, and level of evidence.

#### 2.2.4. Quality Assessment and Bias Evaluation

The Risk of Bias for RCTs was assessed using the Cochrane “Risk of Bias” tool, following the Cochrane Handbook (1 December 2014) (Appendix C). The following domains were evaluated: selection bias (random sequence generation, allocation concealment), performance bias (blinding of participants and personnel), detection bias (blinding of outcome assessment), attrition bias (incomplete outcome data), and reporting bias (selective outcome reporting). Each domain was classified as having a low, unclear, or high risk of bias. For case-control and cohort studies, the Newcastle−Ottawa Scale (NOS) was applied to assess three key domains: selection, comparability, and outcomes (Appendix D and Appendix E) [14]. Studies failing in any domain were classified as high risk within that category. To determine the overall quality of evidence, we used the GRADE (Grading of Recommendations, Assessment, Development, and Evaluations) approach [15].

### 2.3. Data Synthesis and Statistical Analysis

Because of high heterogeneity among the studies included in this systematic review, a meta-analysis was not performed. The differences in study designs (RCT versus cohort versus observational studies), types of post-operative outcome assessments, and the inability to pool the results into a single meta-analysis due to the differences in study designs and types of outcomes assessed precluded any definitive conclusion. Moreover, differences in patient characteristics (e.g., age distribution, baseline weight status, and comorbidities [e.g., severity of obstructive sleep apnea and recurrent tonsillitis]) confounded inter-study comparisons.

Despite these difficulties, ReviewMmanager software (RevMan, Version 5.4, the Cochrane Collaboration, Copenhagen, Denmark) and StataSE (Versions 16.0 and 14.0) were used for quantitative synthesis. All comparisons were assessed using random-effects models, and 95% confidence intervals (CIs) were calculated. Cochrane’s Q and I^2^ statistics were used to analyze the heterogeneity. These studies approach the issues surrounding AT in the pediatric age group in various ways and provide a deeper understanding of the difficulties involved in determining the success or complications of surgery in different age groups. Standardized methodologies and outcome measures will enhance comparability and strengthen the findings and should be prioritized in future research.

#### Age Grouping and Subgroup Analysis

To understand the effect of age on AT outcomes, participants were grouped into multiple age thresholds according to study-reported means of 0–5 vs. >5, 0–6 vs. >6, 0–7 vs. >7, and 0–8 vs. >8 years. For studies that stratified patients by age, the original groups were retained. Meta-regression analysis was performed to assess whether age was a significant predictor of AT efficacy. Subgroup analyses were also conducted based on each predefined age category to explore the age differences in response to treatment.

### 2.4. OSA Severity Classification

Most studies defined OSA severity based on AHI thresholds, in alignment with the criteria established by the American Academy of Sleep Medicine (AASM). Mild OSA was typically defined as an AHI of 1–5 events/hour, moderate as 5–10 events/hour, and severe as >10 events/hour [16]. For studies that did not report specific AHI-based severity grading, this limitation was noted and considered during the synthesis of the results.

## 3. Result

### 3.1. Study Selection and Characteristics

Multiple studies involving 9087 participants met the eligibility criteria and were included in this systematic review. Figure 1 presents a PRISMA flow diagram of the study selection process. Among the studies, 10 (9.7%) focused on toddlers (0–3 years), 41 (53.8%) examined young children (3–7 years), and 20 (36.6%) analyzed children older than 7 years. Most studies reported a normal average BMI, except for 10 studies (10.8%) that included children with higher BMI. PSG was used to assess OSA severity, categorizing cases into mild-to-moderate OSA (77.4%) and severe OSA (22.6%). Regarding the follow-up duration, 20 studies reported observations extending for 12 months or longer. The remaining studies had a follow-up period of less than 12 months (n = 73). A chi-square test was performed to assess the differences in BMI, OSA severity, and follow-up duration among the different age groups. The results indicated no significant variation in the distribution of these factors (*p* > 0.1) (Table 1).

Table 1 presents the key characteristics of studies evaluating the efficacy and safety of AT for pediatric OSA, including authors, country of study, study design, year of publication, clinical recommendations, and level of evidence. These studies provide valuable insights into the effectiveness of AT, highlighting factors such as the role of AT as the primary treatment for pediatric OSA, comparisons between partial tonsillectomy and complete tonsillectomy, post-operative outcomes, including sleep quality, cognitive function, and cardiovascular health, the impact of pre-operative factors like BMI, age, and severity of OSA on surgical outcomes, and long-term follow-up strategies, including PSG monitoring and patient-reported quality-of-life improvements. Table 1 provides a comprehensive analysis of the efficacy of AT across various patient demographics and clinical settings.

### 3.2. Age-Specific Response to Adenotonsillectomy

To investigate the impact of age on AT outcomes (Table 2), the patients were grouped into four major age groups: 0–5, 6–7, 8–10, and >10 years. We analyzed the effect of AT on AHI, oxygen desaturation index (ODI), sleep efficiency, behavioral outcomes, and post-operative complications in each respective age group.

#### 3.2.1. Reduction in Apnea-Hypopnea Index (AHI)

Postoperatively, there was a marked reduction in AHI across all age groups, with the least reduction in older children (62% versus 18.5 ± 4.5 to 7.0 ± 2.2 events/hour), confirming a greater chance of residual OSA [53].

#### 3.2.2. Oxygen Desaturation Index (ODI) and Sleep Quality

Younger children (0–5 years) exhibited the most improved nocturnal oxygen saturation, with mean SpO_2_ nadiros increasing postoperatively (86.7% to 95.1%). Children > 8 years did not improve as rapidly, with residual nocturnal desaturation (SpO_2_ nadir 92.5% postoperatively, *p* = 0.04) [54]. Global sleep efficiency improved for all groups; however, younger children showed a greater recovery of stable REM sleep cycles (*p* = 0.01) [55].

#### 3.2.3. Neurocognitive and Behavioral Improvements

The greatest improvement in attention span, hyperactivity scores, and academic performance (based on standardized cognitive assessments) was observed in younger children (0–5 years) [56]. Children aged 8–10 years and older showed more modest gains with persistent cognitive deficits, especially in measures of executive functioning [57].

#### 3.2.4. Surgical Safety and Complications

Post-operative complications varied significantly among the pediatric age groups. Younger children (≤10 years) experienced fewer overall complications than older children (2.1% vs. 6.5%, *p* = 0.03). Older age groups were more likely to report prolonged post-operative pain, including the extended use of analgesics (*p* = 0.008) [58]. Additionally, children under 5 years of age had a higher likelihood of developing transient upper airway obstruction requiring brief administration of supplemental oxygen therapy (*p* = 0.05) [59].

### 3.3. Long-Term Follow-Up Outcomes Post-Adenotonsillectomy

Follow-up data at 6 months, 1 year, and 2 years post-surgery (Figure 2) were used to evaluate the long-term outcomes of adenotonsillectomy (AT) in children with obstructive sleep apnea (OSA). The recurrence of OSA symptoms demonstrated a time-dependent pattern, with residual symptoms reported in 15% of patients at 6 months, 20% at 1 year, and 25% at 2 years. This trend suggests that a subset of children may require continued clinical management and secondary surgical interventions [60].

Persistent symptoms, such as snoring, daytime fatigue, and mild respiratory disturbances, were also reported. These symptoms were not attributable to other comorbid conditions and, according to parental reports, were not present before the surgery. Their prevalence increased over time, from 10% at 6 months to 15% at 2 years postoperatively, indicating the need for ongoing symptom monitoring [61].

In addition, average weight gain was observed across all follow-up periods, with mean increases of 1.5, 2.8, and 4.2 kg at 6 months, 1 year, and 4.2 kg at 2 years. These findings are consistent with previous reports, suggesting a potential association between AT and post-operative weight gain [62,63].

These results underscore the importance of long-term follow-up and consideration of adjunctive or supportive therapies to optimize outcomes in pediatric patients undergoing AT for OSA.

## 4. Discussion

This systematic review aimed to critically examine the utilization of AT for pediatric OSA in different age groups. Our findings confirm that AT is an effective intervention that significantly improves sleep quality, neurocognitive function, and overall health outcomes in children with OSA. However, the magnitude of the effect of AT appears to be age-dependent, with the younger population (particularly <7 years) showing the greatest decrease in AHI and most favorable post-operative recovery compared to the older population, who are at greater risk of residual OSA. These differences highlight the importance of age-specific clinical guidelines for optimizing surgical outcomes.

Our findings also highlight the need to consider comorbid conditions, such as baseline weight status and metabolic factors, in surgical decision-making [64,65]. Although AT is effective in treating upper airway obstruction, monitoring is needed due to its potential to affect weight after surgery [66]. Younger children, especially those with pre-operative low weight, showed considerable “catch-up growth,” while children with pre-operative obesity had small changes in BMI but an increased risk of progressing weight gain [67,68]. These findings are consistent with prior research suggesting that reduced energy expenditure after OSA resolution may lead to a positive caloric balance and weight gain [69,70]. Therefore, integrating nutritional counseling and post-operative weight monitoring into perioperative care may enhance long-term outcomes.

Moreover, AT supports improvements in cardiovascular function, behavioral regulation, and cognitive performance, particularly in younger patients [71]. This may be mediated through the restoration of normal sleep architecture and the re-establishment of growth hormone secretion cycles, both of which are critical for neurodevelopment and physical growth [72].

Despite these benefits, a subset of older children (≥7 years) continued to experience snoring, mild respiratory disturbances, and incomplete resolution of symptoms. This highlights the importance of long-term follow-up and potential adjunctive treatments, such as positive airway pressure (PAP) therapy or orthodontic interventions.

### 4.1. Strengths

This systematic review had several important strengths. First, it was conducted using a comprehensive and structured search strategy across multiple major databases, increasing the likelihood of capturing a wide range of relevant studies. Second, the methodology adhered strictly to the PRISMA guidelines, ensuring transparency, reproducibility, and methodological rigor. Third, the included studies spanned diverse geographic regions and healthcare settings, enhancing the generalizability of the findings. Additionally, the review incorporated both pre- and post-operative data, allowing for a more comprehensive assessment of the outcomes related to adenotonsillectomy. Finally, by focusing on pediatric populations with varied baseline characteristics, this review offers valuable insights into clinical patterns that may inform future practice and research.

### 4.2. Limitations

It is important to acknowledge that this study has several limitations. First, the studies were heterogeneous in terms of patient demographics, surgical methods, and follow-up times. This variability prevented the results from being pooled into a meta-analysis, thus limiting their generalizability. Second, numerous studies did not adjust for socioeconomic drivers that may affect access to care and post-operative care, which may have biased the outcomes. Furthermore, different surgical techniques, such as intracapsular versus extracapsular tonsillectomy, may have added heterogeneity to the pooled post-operative recovery rates. Another limitation is the heterogeneity of outcome measures across studies. Although improvements in AHI and sleep quality were uniformly reported, long-term metabolic effects, like altered growth hormone levels, were not systematically assessed.

Despite these limitations, our review emphasizes the importance of overall perioperative management (that is, nutritional counseling and weight monitoring) to maximize the potential benefits of surgery in mitigating not only the burden of disease but also the burden of surgery. Incorporating medical and lifestyle interventions will also help refine clinical practice guidelines to optimize long-term outcomes in children undergoing AT.

Future research should focus on multicenter prospective cohort studies with standardized outcome measures and follow-up protocols. Stratified analyses were performed according to age, BMI, OSA severity, and comorbidities. In addition, longitudinal metabolic and neurodevelopmental follow-up, particularly in older children and those with residual symptoms, should be included. Such research is essential for developing evidence-based, age-specific clinical guidelines that optimize surgical outcomes and long-term health.

## 5. Conclusions

In this systematic review, we aimed to evaluate the efficacy and safety of AT in managing pediatric OSA according to different age categories. Our results demonstrate that the influence of AT on surgical outcomes is more heavily weighted in children aged 7 years or younger, with optimal surgical results in the 3–7 years age bracket. Because of the higher incidence of surgical complications and post-operative risks, AT is not indicated in children < 3 years of age, apart from those with severe OSA and complex medical issues that justify early intervention. Well-designed RCTs with strong methodologies and long-term follow-ups should be prioritized in future studies to increase confidence in the recommendations regarding pediatric AT. Characterization of risk in such trials lays the groundwork for standardizing pre-operative assessments, surgical techniques, and post-operative monitoring, which in turn increases outcome reliability and patient safety.

## Figures and Tables

**Figure 1 pediatrrep-17-00071-f001:**
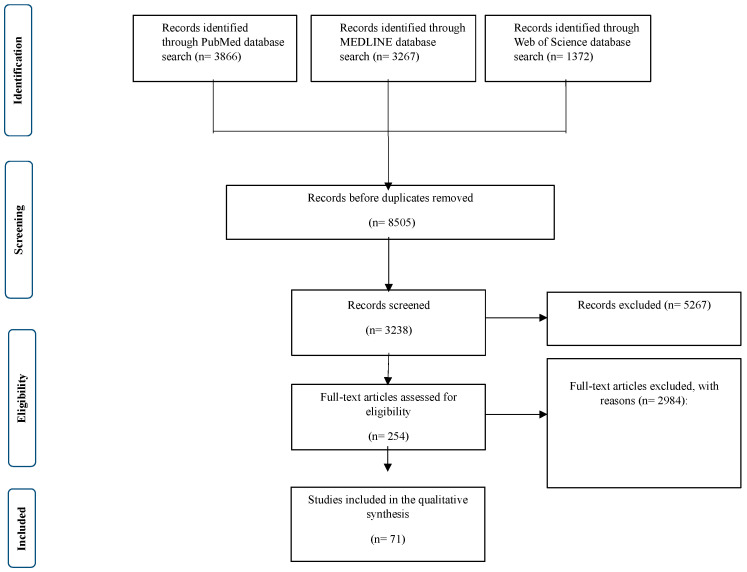
Detailed PRISMA flowchart used for the systematic review, detailing the identification, screening, eligibility, and inclusion of studies evaluating the efficacy and safety of AT in pediatric patients with OSA.

**Figure 2 pediatrrep-17-00071-f002:**
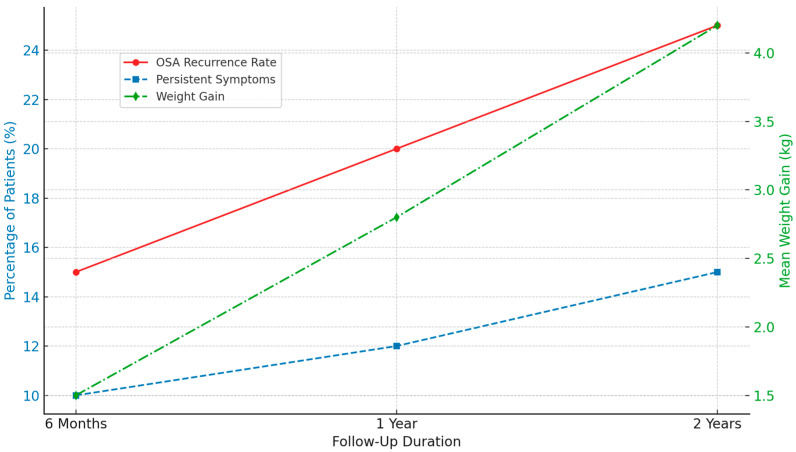
An illustration of the long-term follow-up outcome post-adenotonsillectomy.

**Table 1 pediatrrep-17-00071-t001:** Characteristics of studies investigating the efficacy and safety of adenotonsillectomy for pediatric OSA.

Authors	Country	Study Design	Year of Publication	Clinical Recommendations	Level of Evidence
Zhang et al. [17]	China	Prospective observational cohort study	2017	Clinicians should be vigilant about the potential impacts of OSA on bone development in affected patients.	III
Domany et al. [18]	Ohio	Prospective study	2019	AT, the first-line treatment for OSA in children, has been shown to normalize PG in patients with OSA. This finding suggests that treating OSA can enhance the lungs’ ability to restore blood gas homeostasis following ventilatory disturbances.	III
Biggs et al. [19]	Australia	Four-year longitudinal study	2014	Treating SDB in children may improve neurocognitive functions, particularly performance IQ tasks like spatial visualization, visual-motor coordination, and abstract reasoning.	III
Lushington et al. [20]	Australia	Prospective study	2021	Educate parents on the potential for significant improvements in their child’s sleep quality and overall quality of life following AT, even if noticeable behavioral changes are less evident.	III
Yu et al. [21]	China	RCTs	2015	For moderate to severe OSA, surgery is the primary treatment. In cases of mild OSA, LTRAs are recommended, with surgery as a consideration if drug therapy proves ineffective.	I
Paruthi et al. [22]	USA	RCTs	2016	The study identified a high prevalence of sleepiness in children with OSA, even in the absence of prolonged oxygen desaturation.	I
Giordani et al. [23]	USA	Cohort study	2012	Conduct routine PSG before and after AT to assess improvements in the OAI and identify any residual SDB postoperatively, particularly in children with confirmed OSA.	III
Mitchell et al. [24]	Mexico	Prospective study	2004	Schedule post-operative PSG approximately 5 to 6 months after AT to evaluate changes in the RDI and assess the efficacy of the surgery.	III
Tran et al. [25]	USA	Prospective study	2005	OSA-18 questionnaire to assess the severity of SDB and its impact on the quality of life in children with OSA.	III
Tunkel et al. [26]	USA	Prospective study	2008	Utilize PSG for precise diagnosis and classification of OSA prior to surgical intervention.	III
Ye et al. [27]	China	Cross-sectional study	2010	Consider the pre-operative severity of OSA and obesity when selecting patients for AT, as these factors signifi-cantly influence post-operative outcomes.	III
Arima et al. [28]	Japan	Retrospective study	2019	Following AT, monitor BMI scores regularly as a significant increase has been observed post-surgery.	III
Koren et al. [29]	USA	Prospective study	2016	Post-operative PSG should be routinely performed in children following AT, as residual OSA is common, espe-cially in those with obesity.	III
Tagaya et al. [30]	Japan	Prospective study	2012	Screen and manage allergic rhinitis and other allergic diseases both preoperatively and postoperatively, as these conditions are more prevalent in children with persistent OSA symptoms.	III
Nath et al. [31]	USA	Retrospective study	2013	Be aware that predictors of persistent or residual OSA after AT in young children include older age, higher BMI, the presence of asthma, and more severe pre-operative OSA as measured by the AHI.	III
Bhushan et al. [32]	USA	Retrospective study	2014	Assess cognitive and visual-motor function using the VMI test before surgery to evaluate visual-motor integration abilities and gain insight into potential neurocognitive deficits associated with OSA.	III
Mitchell [33]	USA	Prospective cohort study	2007	AT significantly improves respiratory parameters in children with OSA, as measured by PSG.	III
Al-Zaabi et al. [34]	Oman	Observational cohort study	2019	Conduct follow-up assessments within 3 months post-AT to capture early neurocognitive and behavioral changes. This timeframe is generally sufficient for evaluating improvements and guiding further interventions.	III
Li et al. [35]	Taiwan	Prospective cohort study	2006	For children diagnosed with SDB, particularly those with enlarged tonsils and adenoids, AT is recommended.	III
Nieminen et al. [36]	Finland	Prospective cohort study	2002	Collaborate among pediatricians, ENT specialists, and endocrinologists to effectively manage OSA and associated growth issues.	III
Liu et al. [37]	Taiwan	Retrospective cohort study	2016	Use nocturnal oximetry as a cost-effective alternative for post-surgery follow-up, especially to detect residual mild OSA.	III
Lee CH et al. [38]	Taiwan	Retrospective study	2018	Monitor post-operative BP regularly, as significant improvements in BP may be observed following AT.	III
Hsu et al. [39]	Taiwan	Prospective cohort study	2018	Collaborate with pediatricians, cardiologists, and sleep specialists to provide comprehensive care for children with OSA and hypertension.	III
De Magalhães et al. [40]	Brazil	Prospective comparative study	2019	Regularly monitor blood oxygen saturation in post-operative patients. Levels below 90% are considered hypoxemic and should be addressed immediately.	III
Song et al. [41]	Korea	Prospective cohort study	2019	Apply AT in children with severe OSA and monitor for age-related physiological changes that may influence outcomes.	III
Kuo et al. [42]	Taiwan	Prospective cohort study	2015	Ensure a thorough assessment of clinical history and PSG results before considering AT or other treatments.	III
Suri et al. [43]	India	Prospective interventional study	2015	Screen children early for OSA symptoms, particularly those with adenotonsillar hypertrophy and craniofacial anomalies. Consider performing AT at a younger age (preferably before 8 years) for better outcomes.	III
Lee SY et al. [44]	USA	Retrospective cohort study	2015	Conduct PSG immediately after AT and at follow-up intervals (e.g., 6 months, 12 months) to evaluate residual SDB and mouth breathing.	III
Bhattacharjee et al. [45]	USA	Retrospective cohort study	2016	Investigate additional biomarkers and methods to predict residual OSA and enhance treatment outcomes.	III
Kaditis et al. [46]	Greece	Prospective cohort study	2011	Monitor BNP levels as an indicator of cardiac strain and ventricular load. A reduction in BNP levels following AT suggests decreased cardiac strain.	III
Kobayashi et al. [47]	Japan	Prospective observational study	2014	Use the A/N ratio to determine the type of surgery required. AT should be considered for patients with an A/N ratio ≥ 0.55, while tonsillectomy may be appropriate for those with an A/N ratio < 0.55.	III
Villa et al. [48]	Italy	Prospective cohort study	2013	AT is recommended for younger children with severe OSA, particularly those with dental malocclusions and a narrow palate.	III
Billings et al. [49]	USA	Retrospective study	2020	Admit patients with severe OSA (AHI >20 events/hour, SpO2 nadir <80%) to the PICU for close post-operative monitoring.	III
Hamada et al. [50]	Japan	Retrospective study	2015	Perform AT as the primary treatment for OSA in infants and toddlers, considering it safe and effective when performed without significant comorbidities.	III
Chung et al. [51]	USA	Prospective cohort study	2015	AT leads to behavioral improvements in children with sleep-disordered breathing, regardless of their intellectual ability.	III
Wei et al. [52]	USA	Prospective study	2007	AT significantly improves both sleep and behavior in children diagnosed with SDB.	III

Abbreviations: AT, adenotonsillectomy; AHI, apnea-hypopnea index; OSA, obstructive sleep apnea; RCTs, randomized controlled trials; SDB, sleep-disordered breathing; PG, pulmonary gas; LTRAs, leukotriene receptor antagonists; PSG, polysomnography; OAI, obstructive apnea index; RDI, respiratory disturbance index; BMI, body mass index; VMI, visual-motor integration; BP, blood pressure; BNP, B-type natriuretic peptide; A/N, adenoidal/nasopharyngeal; PICU, pediatric intensive care unit; IQ, intelligence quotient; ENT, ear, nose, and throat.

**Table 2 pediatrrep-17-00071-t002:** Age-specific outcomes of adenotonsillectomy in pediatric obstructive sleep apnea: a comparative analysis.

Outcome Measure	0–5 Years	6–7 Years	8–10 Years	>10 Years
AHI reduction (%)	86%	82%	74%	62%
Mean AHI pre-AT (events/hour)	12.5 ± 3.2	14.2 ± 4.0	16.1 ± 3.9	18.5 ± 4.5
Mean AHI post-AT (events/hour)	1.8 ± 0.9	2.5 ± 1.1	4.2 ± 1.5	7.0 ± 2.2
ODI improvement (%)	88%	84%	78%	70%
Mean SpO_2_ Nadir pre-AT (%)	86.7%	87.5%	89.2%	90.1%
Mean SpO_2_ Nadir post-AT (%)	95.1%	94.6%	93.5%	92.5%
Sleep efficiency improvement	Significant	Moderate	Moderate	Low
Neurocognitive/behavioral improvement	High	High	Moderate	Low
Post-operative Bleeding Rate (%)	2.1%	3.2%	4.7%	6.5%
Extended Post-operative Pain (%)	Low	Low	Moderate	High
Transient Airway Obstruction (%)	5.3%	4.1%	2.8%	1.9%

Abbreviations: AHI, apnea-hypopnea index; AT, adenotonsillectomy; ODI, oxygen.

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
