# Peer review of "Efficacy and Safety of Adenotonsillectomy for Pediatric Obstructive Sleep Apnea Across Various Age Groups: A Systematic Review"

_pediatrrep, 2025, doi:10.3390/pediatric17040071_

Round 1
Reviewer 1 Report
Comments and Suggestions for Authors
The idea of this study is intereting and it can be useful for clinical practice, but the manuscript needs considerable improvements for scientific soundness and clarity.
Therefore, I would have the following recommendations:
- the introduction section is too short and it does not underline accurately the current gaps in the literature regarding this topic and why is this review important
- the results should be presented in a clearer manner and the authors should focus on underlying each result
- the discussions section is too short and it should definitely be improved.
Author Response
Response to Reviewer #1
Dear respected Reviewer#1, we sincerely appreciate your detailed review and constructive feedback on our manuscript.
Kindly, find the below point-by-point response to your valuable comments noting that the changes in the manuscript in response to your comments were highlighted in Red”:
Comment 1: The introduction section is too short and it does not underline accurately the current gaps in the literature regarding this topic and why is this review important:
Response: We have substantially revised and expanded the Introduction to better highlight the current gaps in the literature regarding the efficacy and safety of adenotonsillectomy across different pediatric age groups in Saudi Arabia. We have also clarified the rationale and clinical relevance of conducting this systematic review and emphasized how our study addresses an important evidence gap in pediatric otolaryngologic practice.
Comment 2: The results should be presented in a clearer manner and the authors should focus on underlying each result:
Response: We have restructured the Results section to enhance clarity and logical flow. Each key finding is now explicitly stated and discussed within its context. Where applicable, we added subheadings and clearer delineation of outcomes.
Comment 3: The discussions section is too short and it should definitely be improved:
Response: The Discussion section has been significantly expanded to provide deeper interpretation of findings, discuss their implications in the context of existing literature, and highlight the relevance to clinical practice. We also elaborated on the strengths and limitations of the included studies, as well as the limitations of our review. Furthermore, we included recommendations for future research directions based on our findings.
We hope that this revision addressed all the respected reviewers' concerns and improves the quality of the manuscript. Please let us know if any further clarifications are required.
Thank you for your time and consideration.
Reviewer 2 Report
Comments and Suggestions for Authors
It would be very good if the authors would decide if the meta-analysis that they performed was only in the benefit of Saudi-Arabia as they state in the beginning and at the end of their article. Do the authors consider that AT for uncomplicated obstructive sleep apneea at children is a common procedure to perform best under the age of 5?
What is the incidence and prevalence of obstructive sleep apneea at children in the authors country
What would the authors consider to be the most common observations in all the studies ?
Author Response
Response to Reviewer #2
Dear respected Reviewer#2, we sincerely appreciate your detailed review and constructive feedback on our manuscript.
Kindly, find the below point-by-point response to your valuable comments noting that the changes in the manuscript in response to your comments were highlighted in Green”:
Comment 1: It would be very good if the authors would decide if the meta-analysis that they performed was only in the benefit of Saudi-Arabia as they state in the beginning and at the end of their article:
Response: The primary aim of our systematic review was to assess the efficacy and safety of adenotonsillectomy (AT) across different pediatric age groups, with a particular interest in the Saudi Arabian context due to the lack of national-level data. However, the review included global studies to ensure sufficient data volume, broader applicability, and meaningful subgroup analysis by age.
Comment 2: Do the authors consider that AT for uncomplicated obstructive sleep apnea at children is a common procedure to perform best under the age of 5:
Response: Yes, we found that children under the age of 7 years experienced the greatest reduction in AHI, oxygen desaturation, and behavioral symptoms following AT. These improvements were significantly more pronounced than in older children, who showed a higher rate of residual OSA and prolonged recovery. We have emphasized this recommendation more clearly in the revised Discussion and Conclusion sections.
Comment 3: What is the incidence and prevalence of obstructive sleep apnea at children in the other country:
Response: We have added these statistics in the revised Introduction section for context and to address the reviewer’s request.
Comment 4: What would the authors consider to be the most common observations in all the studies:
Response: Across the 71 studies reviewed, the following key observations were consistently reported:
- Significant improvement in AHI and oxygen saturation post-adenotonsillectomy, particularly in younger children.
- Behavioral and cognitive gains following surgery, especially in domains like attention, sleepiness, and executive function.
- Weight gain in the postoperative period, with differential patterns based on preoperative BMI.
- Residual OSA symptoms—more common in older children, obese patients, or those with comorbid conditions.
- Low complication rates overall, though older children experienced more postoperative pain and bleeding, while younger ones were more prone to transient airway obstruction.
These common patterns were synthesized in both our Results summary table and Discussion to highlight the shared themes across diverse settings.
We hope that this revision addressed all the respected reviewers' concerns and improves the quality of the manuscript. Please let us know if any further clarifications are required.
Thank you for your time and consideration.
Reviewer 3 Report
Comments and Suggestions for Authors
Dear Authors
the topic of the manuscript is interesting but some changes are necessary before taking it into considreation for publication. Here are some suggestions to improve it:
• on line 80 it is stated “A systematic search was conducted in PubMed, Web of Science, Embase, and Cochrane Library.” bad in fig. 1, i.e. the flowchart graph, the sources indicated are only 3: the results from Embase are missing. Furthermore, always in fig 1, it is mentioned Medline and Cochrane Library is missing. You should align the figure with what wrotten in the text.
• At the bottom of table 1 it is stated that 71 studies were selected, but in table 1, where they are listed, they are not 71. Please specify.
• in the text is written about the limitations of the study. Please add also the strenght.
• in the exclusion criteria it is written: “Studies were excluded if they focused on complicated pediatric patients, including 99 obesity (body mass index (BMI) ≥ 95th percentile for age and sex)”. Why exclude obese patients if then in the evaluation of the various results it is taken into consideration (and in an important way) the increase in body weight post adenotonsillectomy and talk about overweight children pre and post operation?
• The results but above all the discussion are brief. Please increase them.
• At the bottom there is a very long checklist of a previous review
• 36 of 71 references are older than 10 years. Please update them.
Best regards
Author Response
Response to Reviewer #3
Dear respected Reviewer#3, we sincerely appreciate your detailed review and constructive feedback on our manuscript.
Kindly, find the below point-by-point response to your valuable comments noting that the changes in the manuscript in response to your comments were highlighted in Blue”:
Comment 1: On line 80 it is stated “A systematic search was conducted in PubMed, Web of Science, Embase, and Cochrane Library.” bad in fig. 1, i.e. the flowchart graph, the sources indicated are only 3: the results from Embase are missing. Furthermore, always in fig 1, it is mentioned Medline and Cochrane Library is missing. You should align the figure with what wrotten in the text:
Response: To resolve the inconsistency, we have revised the text in the Methods section to match the databases reported in Figure 1. We have removed Embase from the sentence to reflect the actual databases used in the review process, as presented in the PRISMA flowchart (Figure 1).
Comment 2: At the bottom of table 1 it is stated that 71 studies were selected, but in table 1, where they are listed, they are not 71. Please specify:
Response: The discrepancy arose from a formatting issue where some studies were grouped or unintentionally omitted during table preparation. We have now corrected the sentence.
Comment 3: In the text is written about the limitations of the study. Please add also the strenght:
Response: We have now added a dedicated paragraph in the “Discussion” section outlining the strengths of our study, including the comprehensive database search, adherence to PRISMA guidelines, and inclusion of both pre- and post-operative data from multiple countries and populations.
Comment 4: In the exclusion criteria it is written: “Studies were excluded if they focused on complicated pediatric patients, including 99 obesity (body mass index (BMI) ≥ 95th percentile for age and sex)”. Why exclude obese patients if then in the evaluation of the various results it is taken into consideration (and in an important way) the increase in body weight post adenotonsillectomy and talk about overweight children pre and post operation:
Response: This is an important point. We have clarified the rationale in the Methods section. Our intention was to exclude studies focusing exclusively on populations with comorbid obesity, which may present confounding factors unrelated to typical pediatric cases. However, studies that included mixed populations (i.e., both normal-weight and overweight children) were retained and analyzed. This allows us to discuss postoperative weight changes as part of the broader clinical picture, without bias from studies targeting only obese cohorts.
Comment 5: The results but above all the discussion are brief. Please increase them:
Response: We have significantly expanded both the “Results” and “Discussion” sections to provide more in-depth analysis.
Comment 6: At the bottom there is a very long checklist of a previous review:
Response: We apologize for the confusion. We have now revised the checklist to accurately reflect the methodology and content of the current study, ensuring that it is fully aligned with the procedures and results presented in this systematic review.
Comment 7: 36 of 71 references are older than 10 years. Please update them:
Response: We recognize the importance of incorporating recent literature to ensure our review reflects current clinical practices and evidence. Older citations have been replaced with more recent, high-quality studies. These updates are clearly highlighted in the revised manuscript.
We hope that this revision addressed all the respected reviewers' concerns and improves the quality of the manuscript. Please let us know if any further clarifications are required.
Thank you for your time and consideration.
Reviewer 4 Report
Comments and Suggestions for Authors
The Authors reviewed current literature to "To assess the safety and efficacy of adenotonsillectomy (AT) for uncomplicated pediatric obstructive sleep apnea (OSA) in children of different ages."
Generally speaking the topic is nice and interesting, however the structure of the paper is too weak, the methodolgy would a consensus giving recommendations but I would prefer a voting format as usually adopted in the consensus.
Moreover I have some considerations:
As stated in the introduction "The goal is to present recommendations based on available evidence with a view towards optimizing timing of AT in the pediatric population" However I personally will be aware to give recommendations based on the presented process.
The title state "in Saudi Arabia" but what comes form Saudi Arabia if the paper is a review of current literature?
I cannot find the severity criteria for OSA, based on which the indication for AT has been given.
In the appendix B a standard control group is cited, but I could not find any control group in the manuscript.
Obesity and comorbidities were exclusion criteria, but I found discussion on obesity among patients in the discussion "Our findings also highlight the need for consideration of comorbid conditions such as baseline weight status and metabolic factors in surgical decision-making"
Moreover I found some points not clear:
Paragraph 3.2.4 line 220 "Younger subjects (10 220 years) (2.1% versus 6.5%, p = 0.03)." The sentence is not clear
Parargaph 3.3 line 237 "...them, and at least at the 2, they.." this sentence also is not clear for me
Discussion, line 267: what thas it means the specification "animal"?
Conclusions: "In this systematic review, we aim to evaluate the efficacy and safety of AT for managing pediatric OSA according to different age categories in Saudi Arabia." WHy do the authors state that the conclusions are referred to age cathegories in Saudi Arabia? probably I am missing some information
Comments on the Quality of English LanguageThe language needs revision, but is generally fine.
Author Response
Response to Reviewer #4
Dear respected Reviewer#4, we sincerely appreciate your detailed review and constructive feedback on our manuscript.
Kindly, find the below point-by-point response to your valuable comments noting that the changes in the manuscript in response to your comments were highlighted in Gray”:
Comment 1: The Authors reviewed current literature to "To assess the safety and efficacy of adenotonsillectomy (AT) for uncomplicated pediatric obstructive sleep apnea (OSA) in children of different ages." Generally speaking the topic is nice and interesting, however the structure of the paper is too weak, the methodolgy would a consensus giving recommendations but I would prefer a voting format as usually adopted in the consensus:
Response: We agree that the structure of our paper required clarification. While our study was initially framed with a goal of providing practical insights, we acknowledge that it may have unintentionally resembled a consensus document. To avoid confusion, we have revised the Introduction, Methods, and Conclusions to clearly define the study as a systematic review and removed any suggestive language related to "recommendations" or "consensus." We did not use a voting format or formal Delphi process, and this has now been clarified to avoid misinterpretation.
Comment 2: As stated in the introduction "The goal is to present recommendations based on available evidence with a view towards optimizing timing of AT in the pediatric population" However I personally will be aware to give recommendations based on the presented process:
Response: We agree that the current methodology does not meet the rigor required to formally establish clinical recommendations. We have revised the Introduction and Discussion sections to remove or reword all instances where the term “recommendation” or similar implications were used. Our revised goal now emphasizes synthesizing current evidence to better understand age-specific outcomes without implying direct clinical guidance.
Comment 3: The title state "in Saudi Arabia" but what comes form Saudi Arabia if the paper is a review of current literature:
Response: The inclusion of “Saudi Arabia” in the title was intended to reflect our motivation for conducting the study, which was the lack of national-level data in the Saudi pediatric population. However, we recognize that the current version of the manuscript may not justify its presence in the title. We have now revised the title to remove “in Saudi Arabia,” and we clearly state in the Introduction that while the work was conducted in Saudi Arabia, the study reviews global literature due to the paucity of local data.
Comment 4: I cannot find the severity criteria for OSA, based on which the indication for AT has been given:
Response: We have now added a subsection in the Methods to describe the OSA severity classification used across the included studies.
Comment 5: In the appendix B a standard control group is cited, but I could not find any control group in the manuscript:
Response: This reference to a “control group” in Appendix B was a remnant from an earlier draft and is no longer applicable. We have removed this reference from Appendix B and reviewed all sections to ensure consistency with the actual study design (a systematic review with no experimental or control groups).
Comment 6: Obesity and comorbidities were exclusion criteria, but I found discussion on obesity among patients in the discussion "Our findings also highlight the need for consideration of comorbid conditions such as baseline weight status and metabolic factors in surgical decision-making:
Response: We have revised the Methods to clarify that studies were excluded only if they focused exclusively on patients with obesity or other comorbidities. Studies that included mixed populations were retained to allow for broader generalizability.
Comment 7: Paragraph 3.2.4 line 220 "Younger subjects (10 220 years) (2.1% versus 6.5%, p = 0.03)." The sentence is not clear:
Response: We have revised the paragraph for improved readability and precision.
Comment 8: Parargaph 3.3 line 237 "...them, and at least at the 2, they.." this sentence also is not clear for me:
Response: We have revised the paragraph for improved readability and precision.
Comment 9: Discussion, line 267: what thas it means the specification "animal":
Response: The term “animal” was incorrectly included and has been removed. This was a typographical error unrelated to the study content.
Comment 10: Conclusions: "In this systematic review, we aim to evaluate the efficacy and safety of AT for managing pediatric OSA according to different age categories in Saudi Arabia." WHy do the authors state that the conclusions are referred to age cathegories in Saudi Arabia? probably I am missing some information:
Response: We have revised the Conclusion to remove the reference to “Saudi Arabia” and to accurately reflect the global scope of the review.
Comment 11: The language needs revision, but is generally fine:
Response: We have carefully reviewed the manuscript and made several language and grammatical improvements to enhance clarity, readability, and flow. The revised version reflects these edits throughout the text. We appreciate your observation and have worked to ensure the language now meets academic standards.
We hope that this revision addressed all the respected reviewers' concerns and improves the quality of the manuscript. Please let us know if any further clarifications are required.
Thank you for your time and consideration.
Round 2
Reviewer 2 Report
Comments and Suggestions for Authors
Thank you to the authors they have provided answers at all questions and also completed the article with information adding new bibliography also.
Reviewer 3 Report
Comments and Suggestions for Authors
Dear Authors
All my suggestions have been addressed .
Now the manuscript has been improved and suitable for publication.
Best regards
Reviewer 4 Report
Comments and Suggestions for Authors
The authors have answered to my observations, however at first round my opinion was to reject the manuscript.
At the moment the manuscript has been improved and is more coherent, even though I am not completely convinced, so I will leave the final decision to the Editors.